# High-Throughput Screening Method Using *Escherichia coli* Keio Mutants for Assessing Primary Damage Mechanism of Antimicrobials

**DOI:** 10.3390/microorganisms12040793

**Published:** 2024-04-14

**Authors:** José A. Martínez-Álvarez, Marcos Vicente-Gómez, Rodolfo García-Contreras, Thomas K. Wood, Fátima Berenice Ramírez Montiel, Naurú Idalia Vargas-Maya, Beatriz Liliana España-Sánchez, Ángeles Rangel-Serrano, Felipe Padilla-Vaca, Bernardo Franco

**Affiliations:** 1Departamento de Biología, División de Ciencias Naturales y Exactas, Universidad de Guanajuato, Noria Alta S/N, Guanajuato 36050, Mexico; 2Departamento de Microbiología y Parasitología, Facultad de Medicina, Universidad Nacional Autónoma de México, Mexico City 04510, Mexico; 3Department of Chemical Engineering, Pennsylvania State University, University Park, PA 16802-4400, USA; 4Centro de Investigación y Desarrollo Tecnológico en Electroquímica CIDETEQ S.C., Parque Tecnológico Querétaro s/n, Sanfandila, Pedro Escobedo, Querétaro 76703, Mexico

**Keywords:** Keio collection, silver nanoparticles, *Escherichia coli*, primary damage, xenobiotic analysis, KatG catalase, bioreporter strains

## Abstract

The *Escherichia coli* Keio mutant collection has been a tool for assessing the role of specific genes and determining their role in *E. coli* physiology and uncovering novel functions. In this work, specific mutants in the DNA repair pathways and oxidative stress response were evaluated to identify the primary targets of silver nanoparticles (NPs) and their mechanism of action. The results presented in this work suggest that NPs mainly target DNA via double-strand breaks and base modifications since the *recA, uvrC, mutL*, and *nfo* mutants rendered the most susceptible phenotype, rather than involving the oxidative stress response. Concomitantly, during the establishment of the control conditions for each mutant, the *katG* and *sodA* mutants showed a hypersensitive phenotype to mitomycin C, an alkylating agent. Thus, we propose that KatG catalase plays a key role as a cellular chaperone, as reported previously for the filamentous fungus *Neurospora crassa*, a large subunit catalase. The Keio collection mutants may also be a key tool for assessing the resistance mechanism to metallic NPs by using their potential to identify novel pathways involved in the resistance to NPs.

## 1. Introduction

The increase in infectious diseases caused by microorganisms has become one of the most serious public health problems worldwide. The irrational use of antibiotics, prolonged regimens, and their use as prophylactic treatment have caused bacteria to develop resistance [1]. To solve this problem, novel approaches and the development of new antimicrobial agents have emerged as alternatives for reducing the new generation of resistant microorganisms [2]. The use of novel agents requires elucidating their action mechanism, and high-throughput screening methods are urgently needed.

The basis for the use of antibiotics dates to ancient human populations that used herbs, mold, and other natural resources to fight infections [3]. The most common targets for antibiotics in bacterial cells are the cell wall, protein translation machinery, and DNA/RNA polymerases. Resistance is linked to genes associated with the repair or bypass of these targets [4].

Among the main resistance mechanisms are the inactivation of the drug by specific enzymes, the modification of the target site of the drug, mutations on the drug target, expression of efflux pumps, and the reduction of cell permeability to reduce drug influx [3]. Furthermore, mobile genetic elements, such as plasmids, encode bona fide resistance genes, some of which are clinically relevant, and the effect of the environment as a selective pressure to enhance the transmission of such elements is important [5,6]. Beyond these, deciphering the remaining physiological cell response mechanisms involved in antibiotic resistance is a fertile field for discovering novel targets for controlling multi-resistant microorganisms.

Overall, drug discovery is complicated due to bottlenecks such as toxicity or low activity in novel molecules. With computational strategies, the discovery of molecules has accelerated, and the bottlenecks have reduced [7]. Another useful approach is the repurposing of known molecules as antimicrobials [8]. Moreover, phages and nanobiotechnology approaches may be the future for containing and limiting the spread of multi-resistant bacteria and other microorganisms [9].

As an example of nanobiotechnology, non-cytotoxic nanoparticles (NPs) with high antibacterial activity have been reported that can be used in dentistry [10]. The report by Skóra and colleagues [10] indicated that nanoparticles can also bypass the effect on normal microbiota, which is a substantial advantage over canonical antimicrobials. Although the development of new nanostructured materials with bactericidal capacity has been the subject of study to determine possible mechanisms of bacterial resistance [11,12], the knowledge of the physiological response in bacteria towards alternative or non-conventional antimicrobials is limited and needed to prevent resistance.

Proteomics has rendered important insights into the cell response to NPs. In *Streptococcus suis*, Liu and colleagues [13] determined, using quantitative proteomics, that upon exposure to NPs, *S. suis* cell-wall proteins were downregulated, and antioxidant proteins were overexpressed and correlated with the production of reactive oxygen species (ROS) in a dose-dependent manner [13]. The effect on the production of ROS of silver NPs has been documented in other systems and seems to be the primary mode of action of silver NPs [9,14]. Nevertheless, using quantitative proteomics to assess the primary antimicrobial mechanism is costly and time-consuming.

Our group has studied the antimicrobial effect of silver NPs synthesized using green chemistry and assessed the physiological response of the bacterial cell using bioreporter strains [15]. We showed that the silver NPs produced cell membrane and DNA damage, and that the plausible mechanism of both damaged structures is the production of ROS [15]. The value of bioreporter strains is that the effect of antimicrobials and nanomaterials may be determined based on primary damage information in vivo [16].

The commercial use of any antimicrobial agent possesses the risk of generating resistance, and it is documented that NPs can elicit resistance in microorganisms [17]. The most important mechanisms of resistance towards NPs involve efflux pumps, electrostatic repulsion of the NPs, biofilm formation, detoxifying enzymes, and genetic changes [17].

Kamat and Kumari [17] reviewed the number of days or generations needed to render a population resistant to NPs and they found that, depending on the composition of the NP, it can result in resistance in as little as 90 min. As an example, Panáček and colleagues [18] reported that exposure to NPs generated resistant subpopulations in as little as eight consecutive culture steps during the exposure to sublethal concentrations of silver NPs, and the mechanism is mediated by flagellin and its aggregative capability. The proposed mechanism seems to be related to the genetic background of the strain, since the authors did not find any genetic change in the genomes of the generated resistant strains [18]. As a countermeasure, using some natural products, such as pomegranate extract, helps reduce resistance [18].

The Keio collection [19] is a set of 3985 singe-deletion mutants of *Escherichia coli*. The Keio collection has been updated [20] and the current collection contains 3864 single-deletion mutants and 25 ORFs were discarded from the collection since there were duplications during the gene interruption [20]. However, the collection has been proven to be useful in assessing gene function and its relationship with the environment or specific phenotypes [21,22,23], and as an example of toxicity evaluations, genome-wide studies for colicin cytotoxicity have been performed [24]. Thus, the Keio collection is a valuable tool for assessing gene–function relationships and extensive studies relating gene loss to phenotypes and sensitivity to antimicrobials.

In this work, we characterized the phenotypic response of Keio collection mutants in DNA repair and in the oxidative stress response pathway genes to the action of commercial silver NPs, analyzed the effect of bona fide damaging agents, such as mitomycin and hydrogen peroxide, and determined the extent of survival using a spot assay in comparison with the Keio collection parental strain (BW25113). Moreover, the simple method developed here shows that the enhanced sensitivity in DNA repair and oxidative stress deletion mutants indicates that the primary action of silver NPs is to induce double-strand breaks in the DNA rather than inducing ROS.

Serendipitously, when conducting the control conditions for the mutants tested, the Δ*katG* mutant showed lethality when exposed to mitomycin C, suggesting that, as described for the fungus *Neurospora crassa*, the large subunit catalase, KatG, may contain chaperone activity in the C-terminal domain, which is needed for survival during exposure to an alkylating agent, such as mitomycin C, which primarily targets DNA but also has the potential for protein modification. Moreover, several bacterial homologs to KatG that contain a similar C-terminal domain may be involved in the resistance to protein-denaturing agents. The results presented here suggest that housekeeping genes may also be involved in the tolerance of or even resistance to antimicrobial agents.

Overall, single-deletion mutants can help assess the primary targets of antimicrobials and pinpoint the pathways more susceptible to damage that may lead to resistance to metallic NPs. Using single-deletion mutants and high-throughput fast assays may benefit the research on the effect of NPs and other antimicrobial agents and companies dedicated to manufacturing diverse antimicrobials, including nanomaterials.

## 2. Materials and Methods

### 2.1. Cell Growth

Cells were grown in Lysogeny Broth [25] at 37 °C for 18 h. Mutant strains were grown in LB media supplemented with kanamycin (50 µg/mL), except when grown in a plate along the parental strain BW25113. DH5α strain was also used as a *recA* mutant reference strain. All the strains used in this study and their genotype are listed in Table 1.

### 2.2. Nanoparticle Preparation

Commercial silver NPs of 20–30 nm diameter, spherical morphology, SSA ~20 m^2^/g, true density 10.5 g/cm^3^, and a purity of 99.5% (product number 0118XH, Skypring Nanomaterials Inc., Houston, TX, USA) were used to determine the microbicidal effect on different mutant backgrounds. First, a stock suspension was prepared by weighing 12,800 µg of silver NPs and resuspending them in 1 mL of PBS-Tween (1%) solution. Afterward, to disperse the NPs, sonication (Fisher Scientific, Waltham, MA, USA) was carried out for two pulses of 60 s with an amplitude of 15.

The dispersed suspension of NPs was used to generate dilutions using PBS-Tween (1%) at the desired final concentration of NPs for each experiment. For each independent experiment, the NPs suspension was prepared at the beginning of the experiment to reduce variability.

### 2.3. Viability Assay

Cells were grown in LB up to a cell density of 10^8^/mL and then placed on a 96-well plate and adjusted to 10^7^/mL in each well, and the NPs suspension was added. For each condition, the diluted NPs were adjusted to add the same volume in all conditions, so that the final cell concentration was 10^6^/mL, and the final volume was 200 µL (adjusted with LB media). The cells and NPs were incubated statically at 37 °C for 1 h. After incubation, from each concentration of NPs, 3 µL was spotted on an LB plate and incubated overnight at 37 °C. After incubation, the plates were analyzed for viability, and results were recorded using a standard document scanner (Hewlett Packard, Hong Kong, China). Minimal inhibitory concentration was assessed to be 200 µg/mL of the commercial silver nanoparticles used using the BW23115 strain (shown as a control).

All assays were conducted in independent triplicate experiments, in all instances in the same plate, and the control condition was included with the parental strain (BW25113). In the figures, a representative result is shown.

### 2.4. Cell Viability Controls

To assess the effect of NPs, the parental and mutant strains were exposed to well-known chemicals that either damage DNA or produce ROS. In this case, mitomycin C and H_2_O_2_ were used to assess the effect of the viability on the parental BW25113 or mutant strains. For the control experiments, the same amount of cells was used (10^6^/mL) and incubated in LB for 60 min, with two concentrations of mitomycin C (2 and 5 µg/mL) or two concentrations of H_2_O_2_ (0.1 and 0.5%). Once the cells were exposed to the desired concentration, 200 µL final volume was placed in a 96-well plate and the cells were serially diluted in the same plate in a 10 factor. From each dilution, 3 µL was spotted in an LB plate and incubated at 37 °C for 18 h. The concentration that reduced in the last dilution of the viability of the BW25113 strain was further used for the mutant strains. In this way, we prevented using a damaging concentration in the parental strain that could not provide a comparable result in the mutant strains.

### 2.5. Protein Model Analysis

The results obtained in this work regarding KatG led us to explore the structural features of this protein. UniProt database (accessed from January to February 2024, https://www.uniprot.org/) was used to retrieve the AlphaFold2 models [26,27,28]. The sequence analysis was performed using Clustal Omega (https://www.ebi.ac.uk/jdispatcher/msa/clustalo) to align the sequences using the default parameters [29]. The sequence alignment was visualized with Alignment Viewer 1.0 (https://alignmentviewer.org/). For the sequence estimates of evolutionary divergence between sequences, the number of amino acid substitutions per site from between sequences is shown. Analyses were conducted using the Equal Input model [30]. All ambiguous positions were removed for each sequence pair (pairwise deletion option). There was a total of 743 positions in the final dataset. Evolutionary analyses were conducted in MEGA11 [31].

Protein structure alignment was conducted with Raptor X DeepAlign software (http://raptorx.uchicago.edu/DeepAlign/submit/) [32,33] using the default settings. Protein structures and alignments were visualized with UCSF-Chimera to highlight the identical regions in structural alignments [34]. Other visualizations were conducted with PyMOL [35]. A rainbow color scheme and cartoon representation were used to assess the N-terminal (blue) and C-terminal (red) end localization.

### 2.6. Reporter Strains

As a proof-of-concept, a previously reported plasmid bearing the *recA* promoter controlling the expression of a chromoprotein (AmilCP) [16] was transformed into BW25113 and Δ*recA* strains to evaluate the sensitivity of the reporter plasmid in the parental and mutant genetic backgrounds. Transformant cells were tested as described previously and transformants were selected in LB-ampicillin (200 µg/mL) [16]. Plates were recorded using a digital camera.

## 3. Results

### 3.1. Keio Mutants Exhibit the Expected Phenotype to Damaging Agents

The use of Keio strains to determine their sensitivity to damaging agents, such as silver NPs, was evaluated for its sensitivity toward known DNA-damaging and ROS agents. In Figure 1, the parental strain was tested for its sensitivity towards mitomycin C and H_2_O_2_ using a spot test (please see Materials and Methods). The control reactions were conducted with 2 and 5 µg/mL of mitomycin C and 0.1 and 0.5% H_2_O_2_ to assess the effect on the parental strain (BW25113), and then we were able to characterize the Keio strains. Four serial dilutions were performed to determine the concentration at which the parental strain was unable to render viable cells at a 10^–4^ dilution versus the control condition that cells were exposed to (PBS-Tween (1%)), which was the dispersion media used for silver NPs testing.

As shown in Figure 1, 5 µg/mL of mitomycin C and 0.1% H_2_O_2_ caused the decrease in cell viability that facilitated the identification of sensitivity towards each compound at a dilution of 10^−4^. The rationale for such a limit was determined in preliminary experiments using Δ*recA* and Δ*katG* mutants, which are highly sensitive to mitomycin C and H_2_O_2_, respectively.

All mutants used in this study were tested for their sensitivity toward mitomycin C and H_2_O_2_ with the above conditions. Figure 2 shows the results for each mutant strain exposed to mitomycin C and H_2_O_2_. As expected, the nucleotide excision repair (*uvrA*, *uvrB* and *uvrC*) mutants all showed a high sensitivity to mitomycin C and less sensitivity to H_2_O_2_. For the base excision repair mutants, the greater mitomycin C sensitivity was observed in the Δ*nfo* mutant but not in the Δ*ung* mutant, and in the *fur* mutant, a higher sensitivity to mitomycin C and H_2_O_2_ was found. The mismatch repair mutants (*mutS*, *mutL*, and *mutY*) behaved as expected: no sensitivity to mitomycin C was found due to their role in mutagenesis coupled with the synthesis of DNA. The recombination mutant Δ*recA* showed the most sensitive phenotype of all mutants tested.

Regarding the mutants in ROS and redox stress response, as expected, the Δ*recA* and Δ*katG* mutants showed the most sensitive phenotype to H_2_O_2_, which is consistent with the hydroxy radical formation that induced DNA double-strand breaks, and the rest of the mutants tested showed a limited sensitivity phenotype toward H_2_O_2_. Unexpectedly, all mutants were sensitive to mitomycin C. After a thorough literature review, we only found a report linking this phenotype to mitomycin C (see below).

Overall, the mutants tested showed the phenotype expected for the type of insult tested here, targeting DNA damage and ROS-induced stress. With the results presented here, we examined the sensitivity of the mutants towards different concentrations of commercial silver NPs.

### 3.2. Silver NPs Generate Double-Strand Breaks in the DNA but No ROS

The experimental setup is described in the Materials and Methods and contemplates the use of dispersed and homogeneous suspensions of silver NPs. Using NPs under these conditions is critical to reproducibility and bypassing the aggregation effect [12] when reducing the NPs’ antimicrobial activity.

For assessing the NPs’ dispersion in PBS/Tween 80 1% at different NP amounts (0.01–100 µg/mL), results were obtained using the dynamic light scattering (DLS) assay, which measures the size distribution profile in suspension (Appendix A). Moreover, representative photographs of the NPs’ stability in PBS buffer, including a series of dispersing agents, where, after 1 h, confirming the use of Tween 80 (1%) allows the NPs’ dispersion (Appendix A). To enrich the commercial NPs characterization, the UV-VIS spectra of NPs and their stability at different times (30–120 min) were measured, confirming the presence of a characteristic peak at 415 nm, indicating the surface plasmon resonance (SPR) of NPs (Appendix A). Finally, the TEM micrograph of a Gram-negative bacteria in contact with 400 µg/mL of NPs indicated the NPs’ adhesion into the bacterial wall and the NPs’ aggregation (Appendix A).

Figure 3 shows the effect of the NP concentrations tested on the parental BW25113 strain, showing almost no effect for the highest concentration tested; however, in the case of the Δ*uvrA*, Δ*uvrB*, Δ*uvrC*, Δ*nfo*, and Δ*ubiE* mutants, the reduction in growth was observed only in the concentrations from 400 to 800 µg/mL of silver NPs. The most substantial effect was observed in the Δ*mutL* and Δ*recA* mutants.

To test a point mutation in RecA, in Appendix A we show the effect of silver NPs on the DH5α strain, showing a hypersensitive phenotype. No effect was observed in the case of oxidative stress and redox response. The most plausible explanation for the greater sensitivity of the Δ*recA* mutant to NPs relative to the Keio mutants deficient in DNA-damage repair is that the tested mutants were single deletions with some remaining DNA repair activity that is eliminated in the Δ*recA* mutant since RecA controls the recognition of the DNA lesion and starts the repair machinery pathway; perhaps in double or triple mutants, the effect can be observed and this is in accordance with the multiple reports showing that metallic (specifically, silver) NPs produce ROS in vivo. To corroborate the phenotype observed in the *recA* Keio mutant, the recA1 mutation was tested in DH5α strain (Appendix A), and in this case, the phenotype also demonstrated higher susceptibility in contact with silver NPs.

### 3.3. Keio Collection Mutants Serves as Chassis for Bioreporter Strains

To expand the use of the Keio mutant collection, we have previously reported the use of a qualitative reporter plasmid that controls the expression of the chromoprotein AmilCP by the *recA* promoter sequence [16]. In this study, the reporter plasmid was transformed into BW25113 and the Δ*recA* strains, and the sensitivity and activation of the reporter were assessed. In Figure 4, the reporter strains were exposed to two different concentrations of mitomycin C. The results show that in the Δ*recA* strain, the activation of the reporter is more evident than in the parental strain (indicated with an arrow) and at a lower concentration (2 µg/mL). The results suggest that a stronger expression of the reporter protein AmilCP can be recorded by eliminating the endogenous gene that responds to the stimulus, or, as in this case, when the damage is more extensive, the signal is stronger.

### 3.4. ΔkatG Phenotype May Be Related to Uncharacterized Function of This Catalase

Keio collection mutants can uncover novel gene functions, previously undetected since using a xenobiotic in certain genetic backgrounds may seem counterintuitive. As stated in previous lines, the *katG* deletion mutant showed an interesting phenotype in mitomycin C, which is not obviously related to the absence of *katG* gene (Figure 2).

To address the effect of mitomycin, we conducted an analysis based on the findings by Hansberg and colleagues and Nava and colleagues [36,37,38] regarding the large subunit catalase (LSC) of *N. crassa*; these authors found that the *N. crassa* LSC C-terminal domain was incorporated into the catalase in the past and is related to the bacterial Hsp31 chaperone, which has been experimentally demonstrated to be a chaperone needed for protein folding. Our results (Figure 2) suggest that KatG in *E. coli* seems to be an ancient form of catalase and chaperone-coupled system, similar to that of the *N. crassa* LSC.

To test this hypothesis, in Figure 5A we compared the sequence (MSA view) of the LSC from *N. crassa* and KatG with the *E. coli* own Hsp31 (HchA) protein. We found it is partially conserved, and the residues contain similar physicochemical properties (hydrophobicity view). The conserved residues found in LSC and KatG in the C-terminal end suggest that the conserved phenylalanine residue shown in [38] is conserved and key for LSC chaperone activity. The last 17 residues in the LSC are needed for chaperone activity, and the authors identified a hydrophobic and charged residue in this region; when mutated using the contrary physicochemical characteristic of each residue, the chaperone activity was also lost. The results suggest that *E. coli* KatG may play a similar role as LSC in *N. crassa*.

In Figure 5B, we provide the evolutionary divergence between sequences, suggesting that both the Hsp31 protein and catalases share the same evolutionary divergence and are closely related. The HchA protein shows more divergence between the two proteins, which is consistent with the sequence alignment between KatG and HchA. The divergence with LSC is also consistent with the putative origin of the C-terminal domain of LSC. To further investigate if this is the case, structural models of KatG, LSC, Bacteroidetes bacterium Hsp31 (UniProt A0A7Y8NWU6, suggested by Hansberg and colleagues [37] as the ancestor of the C-terminal domain in LSC), and the HchA protein were compared. The comparison between KatG and the two Hsp31 homologs showed an overall RMSD of 4.15 (Figure 5C) which is less accurate as reported with the large subunit catalase of *N. crassa* (2.2 Å) [37]. Nevertheless, a similar region is the C-terminal end, which, as reported for LSC, is found in the KatG protein. The comparison with the large subunit catalase with KatG and HchA shows identical regions (Figure 5F), suggesting that the enzymes are conserved and functionally related.

To further support the observation that suggests the conservation of the C-terminal domain, we selected KatG homologs from the UniProt database from phylogenetically distinct organisms. As shown in Figure 6A, using the rainbow color scheme (blue, N-terminal end; red, C-terminal end), the selected examples of catalases annotated as KatG display the same two domain structures as reported for *N. crassa* LSC, which are shown in Figure 6A (PDB 4BIM, monomer extracted with UCSF Chimera). A structural alignment of the selected examples was performed, and as shown in Figure 5B, all bacterial KatG proteins are highly similar (RMSD of 1.58 Å). When all the selected KatG proteins were compared to *N. crassa* LSC, the alignment showed that the structures are highly similar (Figure 6C, RMSD of 3.01 Å). Overall, the protein KatG is highly similar to the fungal LSC, suggesting that the protein has a dual role in eliminating H_2_O_2_ and assisting in misfolded proteins derived from oxidative damage.

## 4. Discussion

In the literature, there are many reports of microbial biosensors for the detection of several contaminants and the cell response associated with exposure to those environmental cues [39,40,41,42,43]. Overall, the reporter strains have robust responses, are inexpensive, and provide important information on the status of pollutants, specifically the bioavailability of the contaminant. Complementary to bioreporters, mutant strains provide relevant information on the sensitivity of specific cell processes. In this report, we provide a framework for using Keio collection mutant strains for the rapid analysis of the effect of silver NPs, showing that double-strand breaks in the DNA are the major antimicrobial effect of NPs and open the avenue for exploring the effect in NP resistance. Serendipitously, the control conditions uncovered that the Δ*katG* mutant strain shows a highly sensitive phenotype to mitomycin C, suggesting that the major catalase KatG is involved in the protein-folding response associated with the C-terminal domain of the enzyme. Along with KatG, the TrxB and SodA proteins may also be involved in protein damage response. This evidence is consistent with the work of Paz and colleagues [44], demonstrating that mitomycin C is a novel thioredoxin reductase inhibitor, which contributes as an anticancer mechanism by affecting the redox cycling in cells. Furthermore, the results presented here and the work of Paz and colleagues [44] suggest that other enzymes containing thiols may be targets of mitomycin C.

The effect of mitomycin C in cell physiology and the toxic effect observed in the antioxidant enzymes can be in part explained by the fact that the effect is not purely directed to DNA, since alkylating agents can add an alkyl group to nucleophilic sites such as sulfhydryl, amino, phosphate, hydroxyl, carboxyl, and imidazole groups [45], which are present not only in DNA but also in RNA and proteins. However, the most studied effect of mitomycin is in the formation of crosslinks in DNA [46].

The results presented here for other mutants in DNA repair synthesis correlate with the expression of RecA upon exposure to mitomycin C and the loss of viability in different mutants such as UvrA, UvrB, and UvrC deficiency [47]. Although mitomycin C affects the expression and total cellular amount of RecA [48], the amount necessary for survival was not determined. However, the role of RecA in generating novel mutations has been reported recently in a phenomenon called hypermutation [49]. Hypermutation is a process related to the SOS response upon DNA damage by inducing the expression of error-prone DNA polymerases and, thus, incorporating mutations in genes associated with antibiotic resistance [49], as shown by Crane and colleagues [49]. This effect happens in vivo but can be inhibited using zinc, which could prevent the further dissemination of resistant bacteria. Additionally, during the hypermutation state, bacterial cells release large amounts of DNA (besides cell elongation). Crane and Catanzaro [50] hypothesized that the released DNA might contribute to the horizontal gene transfer of antibiotic-resistance genes and bacterial clumping and adherence [49]. Mitomycin C was the strongest agent to induce DNA release in these experiments.

Additionally, the phenotype observed in the recA1 mutation in the DH5α strain is interesting since the enzyme is defective in single-stranded DNA-dependent ATPase activity at pH 7.5 but significantly increases at pH 6.2 (Bryant, 1988). In this mutant, only glycine 160 is replaced by an aspartic acid residue [51] and the mutant protein is unable to carry out the three-strand exchange reaction during repair and is strongly inhibited by the single-stranded binding protein and cannot complement a *recA* null mutant [51]. In the context evaluated here, the silver NPs induced large double-strand breaks, which, in the absence of a functional RecA protein in the mutant strain, results in cell death. Using the two different types of mutations confirmed the mechanism of action of silver NPs and the other effects recorded, such as ROS generation, which may result from cell death and not as a primary inhibitory mechanism.

UbiE, an enzyme of the electron transfer machinery, was included in the mutant screening for sensitivity in this work and belongs to a network of genes that facilitate stress-induced mutagenesis (SIM) in *E. coli* K-12 [52]. The SIM links cellular stress with DNA damage and is a network comprising over 93 proteins that are directly or indirectly connected to the mutagenesis mechanism. Al Mamun and colleagues [52] have proposed these genes as therapeutic targets. Here, we showed that the mutant in ubiE is highly sensitive to silver NPs and the control damaging agents. This suggests that it is either an important indicator of cell damage, as Al Mamun and colleagues [52] proposed, or directly linked to DNA damage. Following the same line, Kouzminova and colleagues [53] found that Fur, UbiE, and TopA are proteins needed to prevent chromosome fragmentation in *E. coli* [53]. The result obtained here for the mutant in fur is consistent with the observation that chromosome fragmentation is observed upon damage in the mutant. Here, the silver NPs showed intense cell damage, which is consistent with the lethality observed in the control conditions. The effect reported by Kouzminova and colleagues [53] suggests that UbiE and Fur reduce ROS and thus reduce chromosome fragmentation.

The most surprising result was for the *katG* mutant. In previous studies, the antioxidant enzymes had a profound effect on preventing DNA damage [54,55]. In a previous study, the use of two distinct domains of the KatG protein of *Mycobacterium tuberculosis* rendered protection to DNA-damaging agents in *E. coli* mutants deficient in DNA repair enzymes [55], which is consistent with the observed phenotype in this work. The structural data presented here are consistent with the results shown for *M. tuberculosis* KatG and it remains to be demonstrated that the C-terminal domain constitutes an active chaperone, as demonstrated for the LSC of *N. crassa*. Still, to that end, the construct should consider the integrity of the two domains in *E. coli* KatG. We are currently working on these experiments.

Additionally, a biosensor system has indicated that the transcriptional regulation of KatG is not induced by mitomycin C, which opens the avenue for assessing the regulatory mechanism upon DNA damage and KatG expression [56]. Moreover, the only experimental evidence attributed to the lack of the C-terminal domain in *E. coli* KatG protein is provided by Baker and colleagues [57], who showed that a KatG protein fragment from serine 2 to Aspartate 432 (complete elimination of the C-terminal domain) lacked the catalase activity. In the same study, when the C-terminal domain was expressed and purified separately, it could partially restore the catalase activity [57]. However, Baker and colleagues [57] were not able to assess a mechanistic explanation on why the lack of the C-terminal domain reduced its catalase activity. The finding of the thioredoxin mutant and mitomycin C sensitivity is consistent with the findings of Kumar and colleagues [58], where they demonstrated a direct interaction between TrxA and KatG [58].

Using the mutant strains and a reporter plasmid showed that the response was stronger than in the parental strain (Figure 6). The results presented here allow further engineering in cells deficient in the response gene to enhance their sensitivity. Moreover, we are currently generating a quantitative assay using the reporter protein AmilCP.

As a final observation, the careful maintenance and constant verification of the Keio mutant collection is a significant advantage for using them as bioreporters to assess primary damage in the exposure of xenobiotics. In this work, we did not observe inconsistent results, but routinely conducted strain verification and consistent growing conditions to prevent any anomalies.

The data presented here suggest that mutant strains can render the primary mechanism of action of novel xenobiotic, repurposed, and antimicrobial molecules. Moreover, the mutant strains have an extra value of assessing the role of genes that, in the first instance, seem unrelated to a specific function, as shown here for the antioxidant enzymes. In this work, we exhibited experimental evidence that suggests that KatG has a domain that may be involved in the correct folding of proteins, as shown previously in the *N. crassa* LSC. Further work is being carried out to determine if truncated forms of KatG support growth in the exposure to mitomycin C and exhibit chaperone activity. Furthermore, the Keio collection mutants can be used as a chassis for reporter strains, expanding their use in whole-cell biosensor development.

## Figures and Tables

**Figure 1 microorganisms-12-00793-f001:**
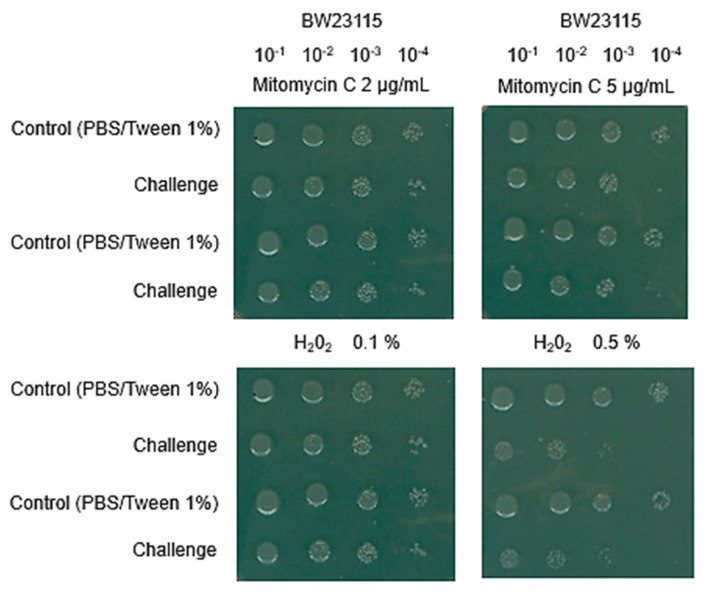
No change in viability in parental strain BW25113 strain and control conditions. To determine the conditions for testing the effect of silver NPs, the Keio collection parental strain, BW25113, was tested against mitomycin C and H_2_O_2_, since the two types of mutant strains tested were either in DNA repair enzymes or antioxidant enzymes. The assays were performed as follows: parental strain was exposed to chemicals that either produced DNA damage or constituted ROS. The same amount of cells was used (10^6^/mL) and incubated (60 min) in LB with two concentrations of mitomycin C (2 and 5 µg/mL) or two concentrations of H_2_O_2_ (0.1 and 0.5%), shown in the upper and lower images. Once the cells were exposed to the desired concentration, 200 µL final volume was placed in a 96-well plate and the cells were serially diluted in the same plate in a 10 factor (indicated in the upper part of the figure). From each dilution, 3 µL was spotted in an LB plate and incubated at 37 °C for 18 h. The non-zero concentration that had no effect on viability of the BW25113 strain was further used for the mutant strains tested in this work. The concentrations that rendered the viability reduction were 5 µg/mL for mitomycin C and 0.1% H_2_O_2_. Here, a duplicate is shown for each condition, but the experiment was carried out as presented three independent times.

**Figure 2 microorganisms-12-00793-f002:**
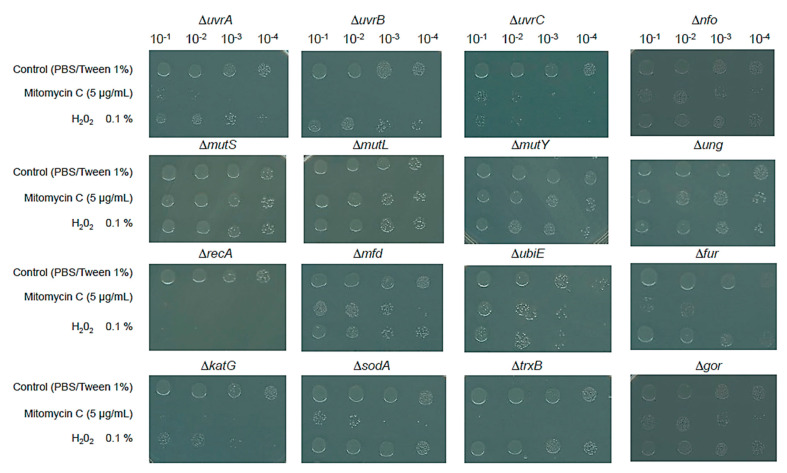
Sensitivity test with control chemicals of the Keio collection mutant set for DNA repair enzymes and antioxidant response enzymes. All mutants were exposed to 5 µg/mL of mitomycin C or 0.1% H_2_O_2_ and viability was determined as described in Figure 1. Each panel shows the result for the control condition (PBS/Tween 1%) and the viability of cells with mitomycin C and H_2_O_2_, indicating on the top the fold dilution as described in Figure 1. The inactivated gene is indicated on top of each panel. A representative result is shown; each experiment was carried out in independent triplicate, including the BW25113 parental strain.

**Figure 3 microorganisms-12-00793-f003:**
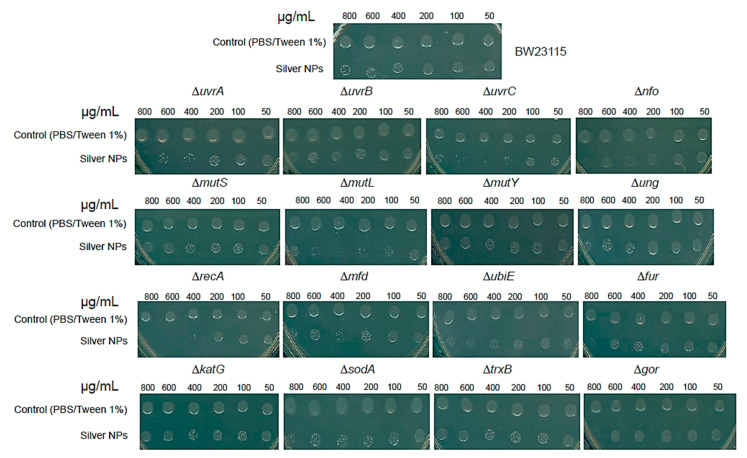
Silver NPs mediate DNA damage through double-strand breaks. Silver NPs were used to challenge all Keio collection mutants tested in this work. Cells were exposed to different concentrations of commercial silver NPs. The concentration of NPs is indicated on top of each panel. The experiment was conducted as described in Materials and Methods, and after incubation with the NPs, 3 µL of cells was spotted on an agar plate. Each experiment included the parental strain BW25113, but we only show the result for each mutant. On top of the figure is shown an example result for the BW25113 strain. Each experiment was conducted in independent triplicate.

**Figure 4 microorganisms-12-00793-f004:**
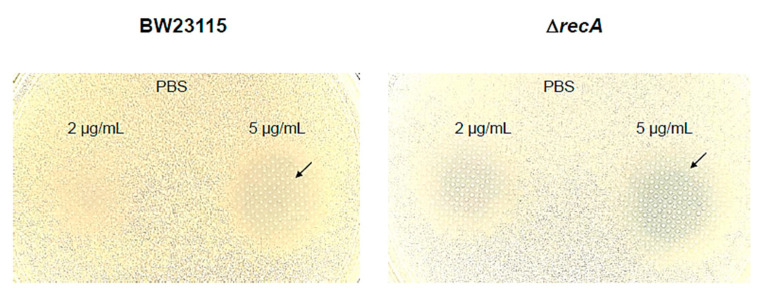
Reporter plasmid using the promoter sequence of *recA* provides a stronger signal in the Δ*recA* mutant strain. A previously reported plasmid (pQE30 plasmid backbone with a replaced promoter sequence for *recA* promoter and expressing AmilCP chromoprotein) was transformed into the parental strain BW25113 and the Δ*recA* mutant strain. In LB agar plates containing ampicillin (200 µg/mL), a lawn of cells was layered using soft LB agar (0.6% agar), and then two concentrations of mitomycin C were spotted on top of the cell lawn. The legends indicate the concentrations of mitomycin C used and the position of a 5 µL PBS control condition. The black arrow indicates the region edge of the halo of AmilCP expression.

**Figure 5 microorganisms-12-00793-f005:**
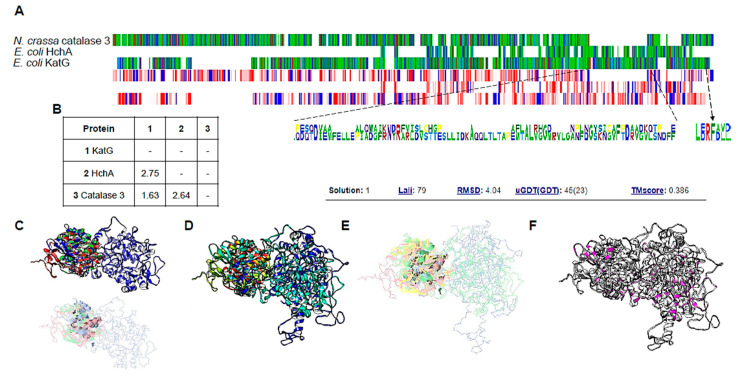
KatG sequence and structural features suggest a role as a chaperone enzyme. Clustal omega alignment between *Neurospora crassa* large subunit catalase (LSC, catalase 3), *Escherichia coli* Hsp31 (HchA), and KatG. Panel (**A**) shows the alignment using Alignment Viewer; the upper view is in the MSA color scheme, and the lower view is in the hydrophobicity color scheme. Overall, the three proteins share similar residues and physicochemical properties. Dashed lines and dashed arrow indicate the region that is only similar between HchA and KatG, and the dashed line indicates the residues located at the C-terminus in LSC and KatG. In panel (**B**), estimates of evolutionary divergence between sequences and the number of amino acid substitutions per site from between sequences are shown. Analyses were conducted using the Equal Input model. All ambiguous positions were removed for each sequence pair. A total of 743 positions were analyzed in the final dataset. Evolutionary analyses were conducted in MEGA11. Panel (**C**) shows the comparison between the AlphaFold2 models of KatG (in blue), HchA (in green), and Bacteroidetes sp. Hsp31 (in red). Structure alignment was conducted with RaptorX, RMSD of 4.15 Å, and a TM score of 0.427. The conserved region is also shown using the same color scheme, and ribbon representation is only shown for the conserved regions. Panel (**D**) shows the structural alignment of KatG (in blue), LSC from *N. crassa* (in cyan), HchA (in red), and Hsp31 from *Bacteroidetes sp*. (in green). Structure alignment was conducted with RaptorX, RMSD of 4.04 Å, and TM score of 0.386. In panel (**E**), the conserved regions of the alignment in panel (**D**) are shown in ribbon. In panel (**F**), using UCSF Chimera, the alignment shown in panel (**D**) was estimated for the identical residues in all structures, which are shown in magenta. KatG model accession number: AF-P13029-F1. HchA model accession number: AF-P31658-F1. LSC PDB 4BIM was used to extract one subunit for comparison.

**Figure 6 microorganisms-12-00793-f006:**
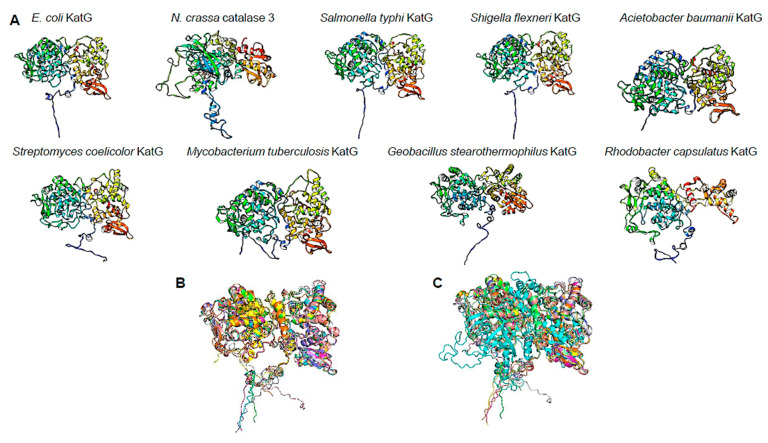
Selected examples of annotated KatG proteins in UniProt database. In panel (**A**), AlphaFold2 models were downloaded from the UniProt database and visualized using UCSF Chimera. All models are presented using the rainbow color scheme, where blue is the N-terminal end and red is the C-terminal end. Accession numbers: *Salmonella typhi* AF-Q8Z303; *Shigella flexneri* AF-Q83IT3; *Acinetobacter baumanii* AF-B0V4K1; *Streptomyces coelicolor* AF-Q9RJH9; *Mycobacterium tuberculosis* AF-P9WIE5; *Geobacillus stearothermophilus* AF-P14412; and *Rhodobacter capsulatus* AF-P37743. In panel (**B**), the structural alignment of all bacterial KatG proteins shown in panel (**A**) is shown. In bright green is *E. coli* KatG, in cyan *Salmonella* KatG, in magenta *S. flexneri*, in yellow *A. baumanii* KatG, in pink *S. coelicolor* KatG, in grey *M. tuberculosis* KatG, in light blue *G. stearothermophilus* KatG, and in orange *R. capsulatus* KatG. RaptorX alignment results: RMSD of 1.58 Å, TM-score of 0.89415. Panel (**C**) shows structural comparison using LSC. Color code: bright green is *E. coli* KatG, in cyan LSC catalase of *N. crassa*, in magenta *Salmonella* KatG, in yellow *S. flexnerii*, in pink *A. baumanii* KatG, in grey *S. coelicolor* KatG, in light blue *M. tuberculosis* KatG, in orange *G. stearothermophilus* KatG, and in light green *R. capsulatus* KatG. RaptorX alignment results: RMSD of 3.01 Å and TM-score of 0.75637.

**Table 1 microorganisms-12-00793-t001:** Genotypes of strains used in this study.

Strain	Genotype *	Antibiotic Resistance
BW25113	Δ(araD-araB)567, ΔlacZ4787(::rrnB-3), λ-, rph-1, Δ(rhaD-rhaB)568, hsdR514	None
JW4019-2	F-, Δ(araD-araB)567, ΔlacZ4787(::rrnB-3), λ-, rph-1, Δ(rhaD-rhaB)568, ΔuvrA753::kan, hsdR514	Kan^r^
JW0762-2	F-, Δ(araD-araB)567, ΔlacZ4787(::rrnB-3), λ-, ΔuvrB751::kan, rph-1, Δ(rhaD-rhaB)568, hsdR514	Kan^r^
JW1898-1	F-, Δ(araD-araB)567, ΔlacZ4787(::rrnB-3), λ-, ΔuvrC759::kan, rph-1, Δ(rhaD-rhaB)568, hsdR514	Kan^r^
JW2146-1	F-, Δ(araD-araB)567, ΔlacZ4787(::rrnB-3), λ-, Δnfo-786::kan, rph-1, Δ(rhaD-rhaB)568, hsdR514	Kan^r^
JW2703-2	F-, Δ(araD-araB)567, ΔlacZ4787(::rrnB-3), λ-, ΔmutS738::kan, rph-1, Δ(rhaD-rhaB)568, hsdR514	Kan^r^
JW4128-1	F-, Δ(araD-araB)567, ΔlacZ4787(::rrnB-3), λ-, rph-1, Δ(rhaD-rhaB)568, ΔmutL720::kan, hsdR514	Kan^r^
JW2928-1	F-, Δ(araD-araB)567, ΔlacZ4787(::rrnB-3), λ-, ΔmutY736::kan, rph-1, Δ(rhaD-rhaB)568, hsdR514	Kan^r^
JW2564-2	F-, Δ(araD-araB)567, ΔlacZ4787(::rrnB-3), λ-, Δung-748::kan, rph-1, Δ(rhaD-rhaB)568, hsdR514	Kan^r^
JW2669-1	F-, Δ(araD-araB)567, ΔlacZ4787(::rrnB-3), λ-, ΔrecA774::kan, rph-1, Δ(rhaD-rhaB)568, hsdR514	Kan^r^
JW1100-1	F-, Δ(araD-araB)567, ΔlacZ4787(::rrnB-3), λ-, Δmfd-739::kan, rph-1, Δ(rhaD-rhaB)568, hsdR514	Kan^r^
JW5581-1	F-, Δ(araD-araB)567, ΔlacZ4787(::rrnB-3), λ-, rph-1, ΔubiE778::kan, Δ(rhaD-rhaB)568, hsdR514	Kan^r^
JW0669-2	F-, Δ(araD-araB)567, ΔlacZ4787(::rrnB-3), Δfur-731::kan, λ-, rph-1, Δ(rhaD-rhaB)568, hsdR514	Kan^r^
JW3914-1	F-, Δ(araD-araB)567, ΔlacZ4787(::rrnB-3), λ-, rph-1, Δ(rhaD-rhaB)568, ΔkatG729::kan, hsdR514	Kan^r^
JW3879-1	F-, Δ(araD-araB)567, ΔlacZ4787(::rrnB-3), λ-, rph-1, Δ(rhaD-rhaB)568, ΔsodA768::kan, hsdR514	Kan^r^
JW0871-1	F-, Δ(araD-araB)567, ΔlacZ4787(::rrnB-3), λ-, ΔtrxB786::kan, rph-1, Δ(rhaD-rhaB)568, hsdR514	Kan^r^
JW3467-1	F-, Δ(araD-araB)567, ΔlacZ4787(::rrnB-3), λ-, Δgor-756::kan, rph-1, Δ(rhaD-rhaB)568, hsdR514	Kan^r^
DH5α (ThermoFisher)	F– φ80lacZΔM15 Δ(lacZYA-argF)U169 recA1 endA1 hsdR17(rK–, mK+) phoA supE44 λ–thi-1 gyrA96 relA1	None

* Underlining indicates the mutated gene on each strain.

## Data Availability

All data are available upon request; all accession numbers are provided in the text.

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
