# Peer review of "High-Throughput Screening Method Using *Escherichia coli* Keio Mutants for Assessing Primary Damage Mechanism of Antimicrobials"

_microorganisms, 2024, doi:10.3390/microorganisms12040793_

Round 1
Reviewer 1 Report
Comments and Suggestions for Authors
This work is intriguing and merits publication. Nonetheless, it would be advantageous to incorporate additional data prior to publishing. Specifically, to better elucidate the efficacy of silver nanoparticle (Ag NP) dispersion, it is recommended to provide fundamental data on the antimicrobial effectiveness of Ag NPs, such as the Minimum Inhibitory Concentration (MIC). Moreover, the chemical composition of the Ag NP dispersion should be detailed as thoroughly as possible, including specifics on the stabilizers utilized. This aspect is crucial as the chemical makeup of the Ag NP dispersion can markedly affect both the outcomes of the tests and how these results are interpreted. Incorporating these details will render the study more exhaustive and enlightening for the audience.
Author Response
Dear reviewer,
In the following lines we address your concerns. Thank you for your positive insight.
This work is intriguing and merits publication. Nonetheless, it would be advantageous to incorporate additional data prior to publishing. Specifically, to better elucidate the efficacy of silver nanoparticle (Ag NP) dispersion, it is recommended to provide fundamental data on the antimicrobial effectiveness of Ag NPs, such as the Minimum Inhibitory Concentration (MIC).
Dear reviewer, thank you so much for the positive insight. Under the conditions tested in this manuscript, the MIC for the commercial silver nanoparticles used is 200 µg/mL in an exposure of 60 min. The BW23115 strain was used for standardization since the observed phenotype of strain DH5a was more sensitive. Also, the BW23115 controls were included in every experiment to ensure the proper preparation of NPs. We have indicated this aspect in the methods section in lines 160-162.
Moreover, the chemical composition of the Ag NP dispersion should be detailed as thoroughly as possible, including specifics on the stabilizers utilized. This aspect is crucial as the chemical makeup of the Ag NP dispersion can markedly affect both the outcomes of the tests and how these results are interpreted. Incorporating these details will render the study more exhaustive and enlightening for the audience.
Thank you for your insight, we have added in the methods section the product number of the AgNPs used (lines 141-142) since the AgNPs used were from a commercial source, and the requested information as supplementary material and indicated in lines 271-281 we refer to Supplementary Figure 1. In Supplementary Figure 1 we included the analysis at different concentration of NPs using Dynamic Light Scattering (up to 100 µg/mL of NPs), the effect of different dispersing agents, showing that Tween 80 is appropriate for the assays shown here. Complementary, we show that using UV-Vis spectra the NPs are stable and finally, TEM analysis shows adhesion with 400 µg/mL NPs bound to the cells, confirming that under the conditions tested the NPs bind to bacterial cells.
We hope this information is sufficient to support the findings of this paper.
-Once again, thank you for the positive assessment of this manuscript.
Reviewer 2 Report
Comments and Suggestions for Authors
In this manuscript Jose A Martinez et al. provide the important findings such as primary damage mechanisms and identifying primary targets in E.coli using silver nanoparticles by studying DNA repair and oxidative stress response pathways . This approach of identifying genes and targeting these genes would provide a better therapeutic approach to treat drug resistance infections.
The manuscript experimental design and the conclusions are appropriate. I believe that the findings of the manuscript are of sufficient novelty and breadth to merit publication in microorganisms journal.
Below are suggestions for major revision of the manuscript:
Sentences are too long in the introduction section, I would like to recommend split the sentences in introduction and rewrite.
Starting from line 68 to 72 this sentence is not written clearly this should be rephrased to understand better.
Few more sentences to be corrected.
Starting from line 76 to 80 is not clear as well, this sentence also should be considered to rewrite and make it clear.
Another statement which is too long starting from line 113 to 118.
Overall, the introduction part would have been better if it is described more precisely.
Results and discussion section should be separated.
IF the authors would like to keep it together , it is fine but it should be more clear discussing results appropriately, rather than deviating so much from the results.
Experimental design and results are great!
This article writing representing more of like a review article rather than research article.
Results section should be more precise to the experimental results and concussions.
Finally, I would strongly recommend to re write the whole results and discussion section more precise and accurate.
I have no major concerns with this article to publish in microorganisms Journal.
Comments on the Quality of English LanguageThis article writing representing more of like a review article rather than research article.
Results section should be more precise to the experimental results and concussions.
moderate editing is required .
Author Response
Dear reviewer,
In the following lines we address your concerns. Thank you for your positive insight.
Suggestions for Authors
In this manuscript Jose A Martinez et al. provide the important findings such as primary damage mechanisms and identifying primary targets in E. coli using silver nanoparticles by studying DNA repair and oxidative stress response pathways. This approach of identifying genes and targeting these genes would provide a better therapeutic approach to treat drug resistance infections.
Dear reviewer, thank you so much for the positive insight of the manuscript. In the following lines we provide the response to the specific issues identified in the manuscript. Again, thank you so much for the positive appreciation of this work.
The manuscript experimental design and the conclusions are appropriate. I believe that the findings of the manuscript are of sufficient novelty and breadth to merit publication in microorganisms journal.
Thank you so much for the positive insight.
Below are suggestions for major revision of the manuscript:
Sentences are too long in the introduction section, I would like to recommend split the sentences in introduction and rewrite.
According to your suggestion and the specific lines to rewrite, we have attended this observation. Thank you for the suggestions.
Starting from line 68 to 72 this sentence is not written clearly this should be rephrased to understand better.
Thank you so much for the observation, we have corrected these lines (68-76).
Few more sentences to be corrected.
Starting from line 76 to 80 is not clear as well, this sentence also should be considered to rewrite and make it clear.
Thank you for the observation, we have corrected this sentence (Lines 77-83).
Another statement which is too long starting from line 113 to 118.
We have split these sentences in two (lines 107-114 and lines 115-123) and revised the writing. Thank you for pointing this out.
Overall, the introduction part would have been better if it is described more precisely.
We apologize with the reviewer, but we do not comprehend to what extent the introduction was not precise. We selected some recent examples of the available literature to introduce the topic in this paper. We are concerned with the possible mechanisms of resistance to nanomaterials and this paper contributes to the use of Keio mutants to evaluate in a future if specific genetic backgrounds could be associated with an increase in resistance. Also, is a method for assessing sensitivity towards nanoparticles in strains defective in specific pathways. We have done writing corrections in this section; we hope they make the introduction clearer.
Results and discussion section should be separated.
IF the authors would like to keep it together, it is fine but it should be more clear discussing results appropriately, rather than deviating so much from the results.
Experimental design and results are great!
Thank you!
This article writing representing more of like a review article rather than research article.
Results section should be more precise to the experimental results and conclusions.
Finally, I would strongly recommend to re write the whole results and discussion section more precise and accurate.
Thank you for the suggestion. We have separated these sections and added subsections in the Results section to provide a better division between results. Also, the comment that the manuscript seems to be a review paper is to reinforce the theoretical data provided we kindly request to maintain the strong literature support for the KatG structural analysis; since we strongly believe that the literature review makes a strong case for the secondary function of KatG. Thank you for the insight.
I have no major concerns with this article to publish in microorganisms Journal.
Once again, thank you for the suggestions and positive insight.
Round 2
Reviewer 1 Report
Comments and Suggestions for Authors
The authors have revised and appropriately addressed the reviewers' comments and revisions, and have agreed to recommend it for publication.